# Metformin abrogates pathological TNF-α-producing B cells through mTOR-dependent metabolic reprogramming in polycystic ovary syndrome

Na Xiao[1,2,3], Jie Wang[2,3], Ting Wang[1], Xingliang Xiong[1], Junyi Zhou[4], Xian Su[2,3], Jing Peng[2,3], Chao Yang[2,3], Xiaofeng Li[5], Ge Lin[1,2,3,5,6], Guangxiu Lu[1,2,3,5,6], Fei Gong[1,5,6]*, Lamei Cheng[1,2,3,4,6]*

[1]Institute of Reproductive and Stem Cell Engineering, School of Basic Medical Science, Central South University, Changsha, China; [2]National Engineering and Research Center of Human Stem Cells, Changsha, China; [3]Hunan Guangxiu Hi-tech Life Technology Co. Ltd, Changsha, China; [4]Guangxiu Hospital, Hunan Normal University, Changsha, China; [5]Clinical Research Center for Reproduction and Genetics in Hunan Province, Reproductive and Genetic Hospital of CITIC-Xiangya, Changsha, China; [6]NHC Key Laboratory of Human Stem Cell and Reproductive Engineering, Central South University, Changsha, China

**Abstract** B cells contribute to the pathogenesis of polycystic ovary syndrome (PCOS). Clinically, metformin is used to treat PCOS, but it is unclear whether metformin exerts its therapeutic effect by regulating B cells. Here, we showed that the expression level of tumor necrosis factor-alpha (TNF-α) in peripheral blood B cells from PCOS patients was increased. Metformin used in vitro and in vivo was able to reduce the production of TNF-α in B cells from PCOS patients. Administration of metformin improved mouse PCOS phenotypes induced by dehydroepiandrosterone (DHEA) and also inhibited TNF-α expression in splenic B cells. Furthermore, metformin induced metabolic reprogramming of B cells in PCOS patients, including the alteration in mitochondrial morphology, the decrease in mitochondrial membrane potential, Reactive Oxygen Species (ROS) production and glucose uptake. In DHEA-induced mouse PCOS model, metformin altered metabolic intermediates in splenic B cells. Moreover, the inhibition of TNF-α expression and metabolic reprogramming in B cells of PCOS patients and mouse model by metformin were associated with decreased mTOR phosphorylation. Together, TNF-α-producing B cells are involved in the pathogenesis of PCOS, and metformin inhibits mTOR phosphorylation and affects metabolic reprogramming, thereby inhibiting TNF-α expression in B cells, which may be a new mechanism of metformin in the treatment of PCOS.

*For correspondence:
gongfei0218@hotmail.com (FG);
LameiCheng@csu.edu.cn (LC)

## Editor's evaluation

This study confirms that TNF-α is increased in peripheral blood B cells from polycystic ovary syndrome (PCOS) and metformin decreases production. The study further suggests potential mechanisms for the increase in TNF-α and reduction due to metformin. This is demonstrated in humans as well as in a mouse model of PCOS. Overall, this is a well-designed study demonstrating the impact of metformin on immune function in PCOS.

## Introduction

Polycystic ovary syndrome (PCOS) is one of the most common endocrine-reproductive-metabolic disorder characterized by hyperandrogenism, polycystic ovaries, and chronic oligo-/anovulation, which can lead to infertility (*Rotterdam ESHRE/ASRM-Sponsored PCOS consensus workshop group, 2004*). Accumulating evidence indicates that PCOS occurs under chronic inflammation, leading to ovarian dysfunction and metabolic disorders (*Hu et al., 2020*). Such inflammation is characterized by elevated levels of inflammatory cytokines (e.g. tumor necrosis factor-alpha [TNF-α], interleukin-6 [IL-6], interferon-γ [IFN-γ]) in the serum, follicular fluid, and ovary (*Qin et al., 2016*; *Xiong et al., 2011*). Published results have reported that CD3⁺CD4⁺ T cells of follicular fluid produce high levels of IFN-γ and IL-2 in women with PCOS and splenic macrophages release more TNF-α in rats with PCOS (*Figueroa et al., 2012*; *Qin et al., 2016*). In chronic inflammatory and autoimmune diseases, such as diabetes mellitus, multiple sclerosis, and systemic lupus erythematosus (SLE), pathologic cytokine-producing B cells have been verified to play important roles in these diseases (*Arkatkar et al., 2017*; *Jagannathan et al., 2010*; *Li et al., 2015*). In a mouse model of SLE, IL-6 production by B cells drives autoimmune germinal center formation and accelerates disease progression (*Arkatkar et al., 2017*). B cells expressing IFN-γ inhibit Treg cell differentiation and exacerbate autoimmune experimental arthritis (*Olalekan et al., 2015*). Our previous studies demonstrated that the peripheral proportion and activity of CD19⁺ B cells were increased in women with PCOS and dehydroepiandrosterone (DHEA)-induced morphological changes to mouse ovaries were prevented by CD19⁺ B cell depletion (*Xiao et al., 2019*), demonstrating that CD19⁺ B cells contribute to the pathogenesis of PCOS. However, the pathogenic role of B cells in PCOS remains unclear and little is currently known about the inflammatory cytokines expression by B cells from PCOS.

Metformin is a widely used oral medication to treat type 2 diabetes (T2D) and also involved in other pharmacological actions, including antitumor effect, antiaging effect, and neuroprotective effect (*Bai and Chen, 2021*; *Bharath et al., 2020*). Published results have suggested that metformin not only reduces chronic inflammation through reducing hyperglycemia and increasing insulin sensitivity, but it also has direct anti-inflammatory effects. For example, it has been shown that metformin reduces Th17 cytokine production in older subjects (*Bharath et al., 2020*), decreases the proinflammatory late/exhausted memory B cell subset, and B cell TNF-α mRNA expression in obesity and T2D patients (*Diaz et al., 2017*). The effects of metformin are thought to be mediated primarily through regulation of the activity of adenosine monophosphate-activated protein kinase (AMPK), the mechanistic target of rapamycin (mTOR), and phosphatidylinositol 3-kinase (PI3K), which are major regulators of metabolic stress responses. Metformin suppresses systemic autoimmunity in Roquin^{san/san} mice through inhibiting B cell differentiation via regulation of AMPK/mTOR signaling (*Lee et al., 2017*). Metformin inhibits the proliferation of rheumatoid arthritis fibroblast-like synoviocytes via inhibiting mTOR phosphorylation through PI3K/AKT signaling (*Chen et al., 2019*). Meanwhile, metformin exerts inflammation-inhibitory effects by altering intracellular metabolic processes. Recent evidence has suggested that metformin alleviates the aging-associated Th17 inflammation by improving mitochondrial bioenergetics, and reducing oxidative stress via lowering ROS production (*Bharath et al., 2020*). At present, metformin is also widely used among women with PCOS and reduces insulin resistance, increases ovulation and improves clinical pregnancy outcomes (*Morley et al., 2017*). We hypothesize that pathological B cells may exist in PCOS patients, and metformin may alleviate PCOS symptoms by regulating the activity of pathological B cells.

Here, we showed for the first time that the increase of pathological B cells producing TNF-α was associated with PCOS. Peripheral blood (PB) B cells from women with PCOS produced higher levels of TNF-α, and the percentage of TNF-α⁺ cells in B cells was positively associated with serum AMH levels. However, the expression of TNF-α in B cells was reduced after oral administration of metformin in women with PCOS, and metformin had a similar inhibitory effect on TNF-α expression in B cells in vitro. In DHEA-induced mouse PCOS model, metformin improved PCOS phenotypes, accompanied with inhibition of TNF-α expression in splenic B cells. We further demonstrated that metformin induced mitochondrial remodeling, reduced glucose uptake in B cells from women with PCOS, and altered metabolic intermediates in splenic B cells from DHEA-induced PCOS mice. The inhibition of TNF-α expression and metabolic reprogramming in B cells of PCOS patients and mouse model by metformin were related to the decrease of mTOR phosphorylation. Together, these results identify a new mechanism that TNF-α-producing B cells are involved in the pathological process of PCOS,

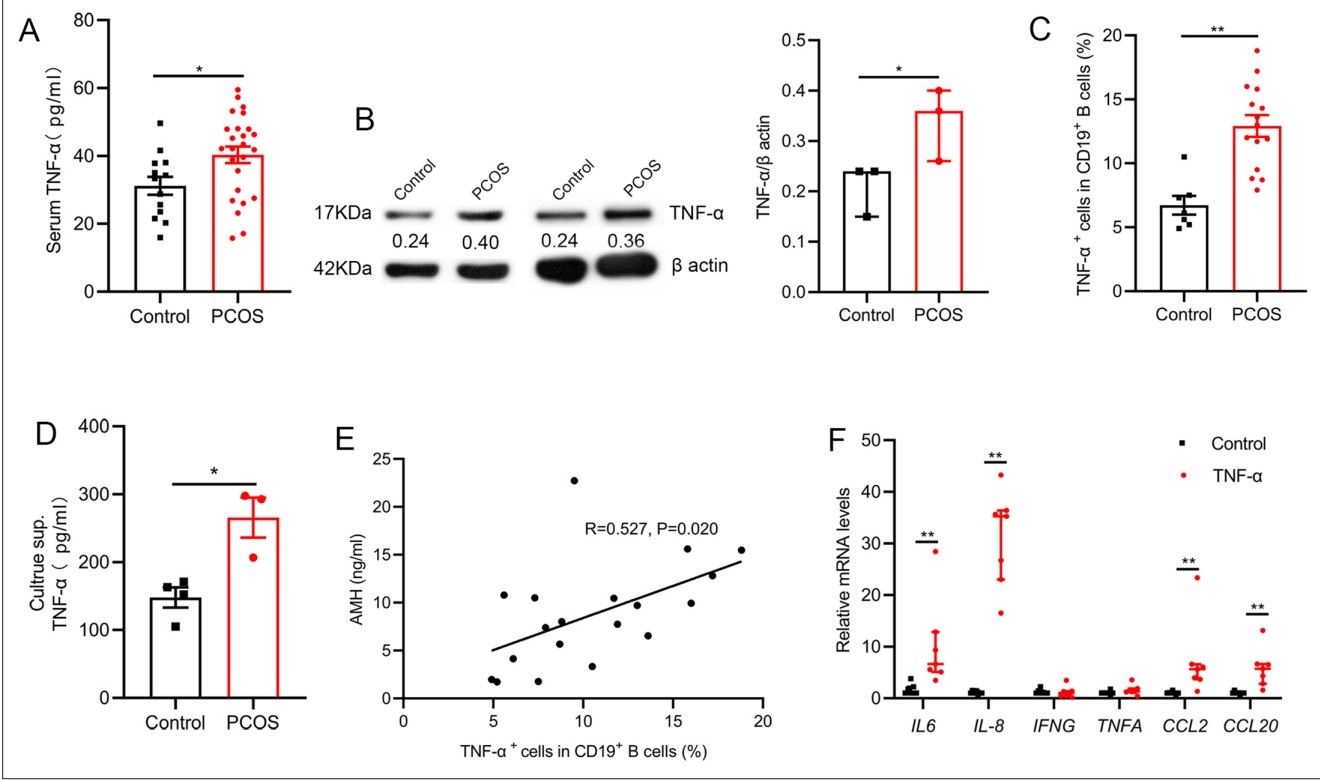

**Figure 1.** Tumor necrosis factor-alpha (TNF-α) production by pathological B cells in women with polycystic ovary syndrome (PCOS). (**A**) Serum TNF-α concentration was assessed with multiplex assays (n=13 [control] and 25 [PCOS]). (**B**) TNF-α expression in CD19+ B cells was determined by Western blot. Shown are representative images and quantification analysis of the ratio of TNF-α/β actin (n=3 per group). (**C**) Percentage of TNF-α+ cells in CD19+ B cells from peripheral blood (PB) stimulated with CpG, CD40L plus anti-IgM/IgA/IgG (n=7 [control] and 15 [PCOS]) was assessed by flow cytometric analysis. (**D**) TNF-α in the culture supernatants (sup.) was assessed by ELISA (n=3–4 per group). (**E**) The correlation between the percentage of TNF-α+ cells in CD19+ B cells and serum AMH levels by Pearson's correlation analyses (n=19). (F) Interleukin-6 (*IL-6*), *IL-8*, *IFNG*, *TNFA*, *CCL2*, and *CCL20* mRNAs expression levels in granulosa cells were measured by qPCR (n=7 per group). For (A), (C), and (D), p values were determined by two-tailed Student's t-test and data are presented as means ± SEM. For (B) and (F), p values were determined by two-tailed Mann-Whitney U-test and data are presented as medians with interquartile ranges. *p<0.05; **p<0.01.

The online version of this article includes the following source data for figure 1:

**Source data 1.** Data points for graphs in *Figure 1*.

and metformin may inhibit TNF-α expression of pathological B cells via mTOR-dependent metabolic reprogramming.

## Results

### TNF-α production by pathological B cells in women with PCOS

Inflammation plays an essential role in the pathogenesis of PCOS. To define PCOS-associated cytokine profiles, we first quantified the levels of serum inflammatory cytokines in 25 women with PCOS and 13 age-matched control subjects. Results showed that the level of TNF-α was higher in women with PCOS (*Figure 1A*). Our previous report demonstrated the proportion and activity of CD19+ B cells were significantly increased in women with PCOS (*Xiao et al., 2019*), thus we questioned whether the abnormal B cells are responsible for the increase of serum TNF-α in women with PCOS. The results showed that B cells isolated from PB of women with PCOS expressed higher amounts of TNF-α protein than those from the controls (*Figure 1B*). We also quantified TNF-α expression in B cells after stimulation with B cell-targeted activator (CpG, CD40L plus anti-IgM/IgA/IgG) by flow cytometry and found that the percentage of TNF-α+ cells in CD19+ B cells was notably higher in women with PCOS than that in control subjects (*Figure 1C*). More TNF-α was secreted into the culture supernatants (*Figure 1D*). Anti-Müllerian Hormone (AMH) is solely secreted by granulosa cells (GCs) of the pre-antral and small

antral ovarian follicles, and could be an effective indicator for the diagnosis of PCOS (*Teede et al., 2019*). We analyzed the relationship between the percentage of TNF-α+ cells in B cells and the level of serum AMH by Pearson's correlation analysis, and the results showed that the percentage of TNF-α+ cells in CD19+ B cells was positively associated with serum AMH levels (*Figure 1E*). To further investigate the possible role of TNF-α in GCs, luteinized GCs from healthy individuals were isolated, and cultured in RPMI-1640 containing 10% FBS with or without TNF-α. We found that TNF-α could significantly increase the expression of *IL-6*, *IL-8*, *CCL2*, and *CCL20* mRNAs in GCs (*Figure 1F*). Together, these data indicate that the increase of pathological B cells producing TNF-α is associated with PCOS.

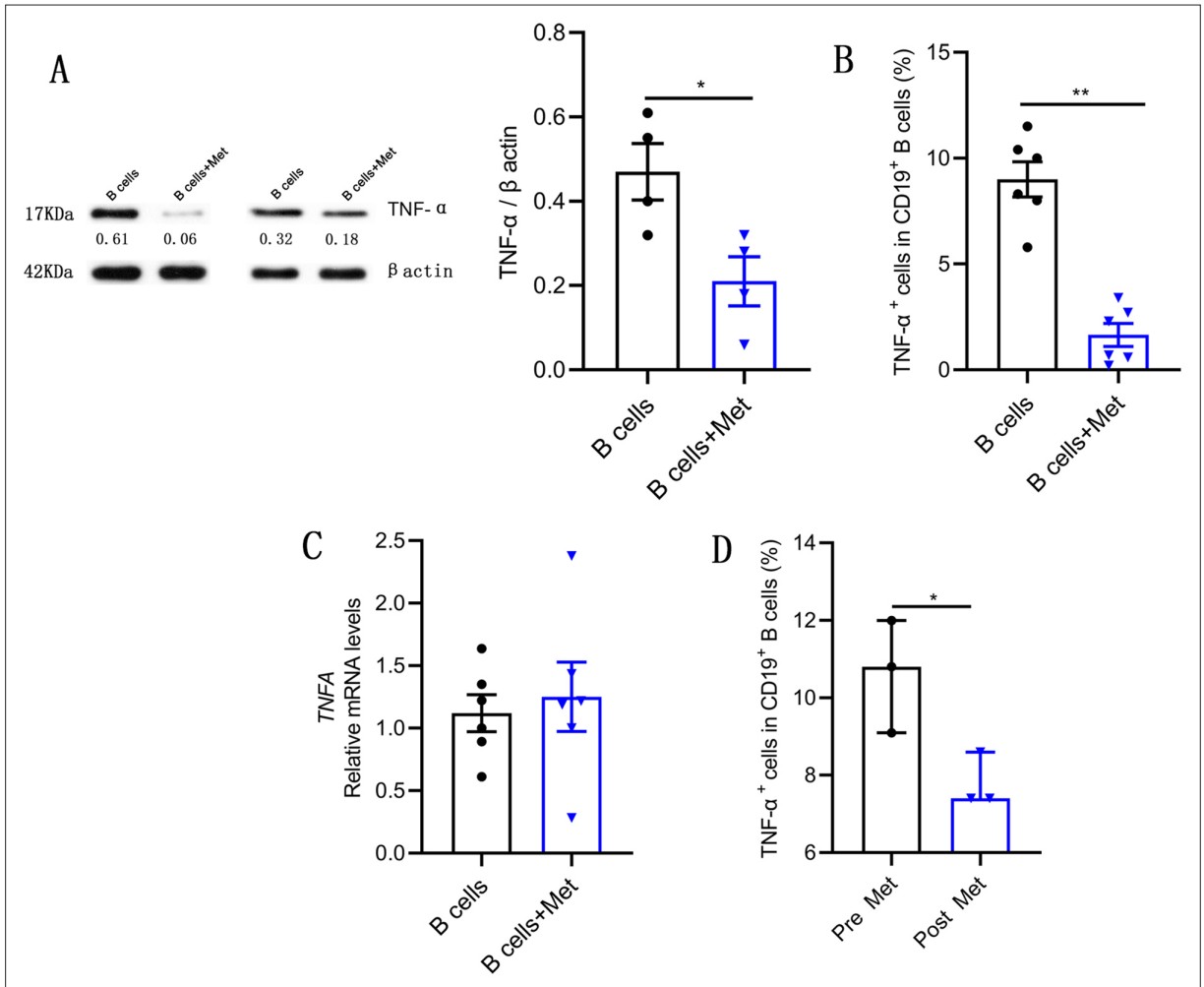

**Figure 2.** Metformin inhibits tumor necrosis factor-alpha (TNF-α) production by pathological B cells in polycystic ovary syndrome (PCOS). (**A–C**) B cells isolated from peripheral blood (PB) of women with PCOS were cultured in RPMI-1640 medium supplemented with B cell-targeted activator (CD40L, CpG plus anti-IgM/IgA/IgG) with or without metformin (12.5 mM, Met) for 48 hr in vitro. (**A**) TNF-α expression in CD19+ B cells by Western blot. Shown are representative images and quantification analysis of the ratio of TNF-α/β actin (n=4 per group). (**B**) Percentage of TNF-α+ cells in CD19+ B cells by flow cytometric analysis (n=6 per group). (**C**) *TNFA* mRNA expression levels in B cells (n=6 per group). (**D**) Percentage of TNF-α+ cells in CD19+ B cells from women with PCOS, before (pre) and after (post) treatment with metformin for 1 month (n=3 per group). For (A), (B), and (C), p values were determined by two-tailed Student's t-test and data are presented as means ± SEM. For (D), p values were determined by two-tailed Mann-Whitney U-test and data are presented as medians with interquartile ranges. *p<0.05; **p<0.01.

The online version of this article includes the following source data and figure supplement(s) for figure 2:

**Figure supplement 1.** Metformin does not affect B cell proliferation and apoptosis.

**Source data 1.** Data points for graphs in *Figure 2* and its supplements.

## Metformin inhibits TNF-α production in pathological B cells

The glycemic control drug metformin reduces B cell intrinsic inflammation in individuals with obesity and T2D and is widely used among women with PCOS (*Diaz et al., 2017*; *Morley et al., 2017*). To study the effect of metformin on the TNF-α production in pathological B cells, B cells isolated from PB of women with PCOS were cultured in RPMI-1640 medium supplemented with B cell-targeted activator with or without metformin. The expression of TNF-α in B cells was analyzed by Western blot analysis and flow cytometry. The results showed that metformin significantly inhibited TNF-α production in PCOS-derived B cells (*Figure 2A–B*), but did not affect proliferation and apoptosis of B cells (*Figure 2—figure supplement 1*). These changes occurred post-transcriptionally because there was no significant change in *TNFA* mRNA levels (*Figure 2C*). Furthermore, we analyzed the expression level of TNF-α in pathological B cells before and after oral metformin for 1 month in PCOS patients. In line with the in vitro results, the treatment of metformin significantly diminished the expression of TNF-α in B cells (*Figure 2D*). These results demonstrate that metformin inhibits TNF-α production in pathological B cells from women with PCOS.

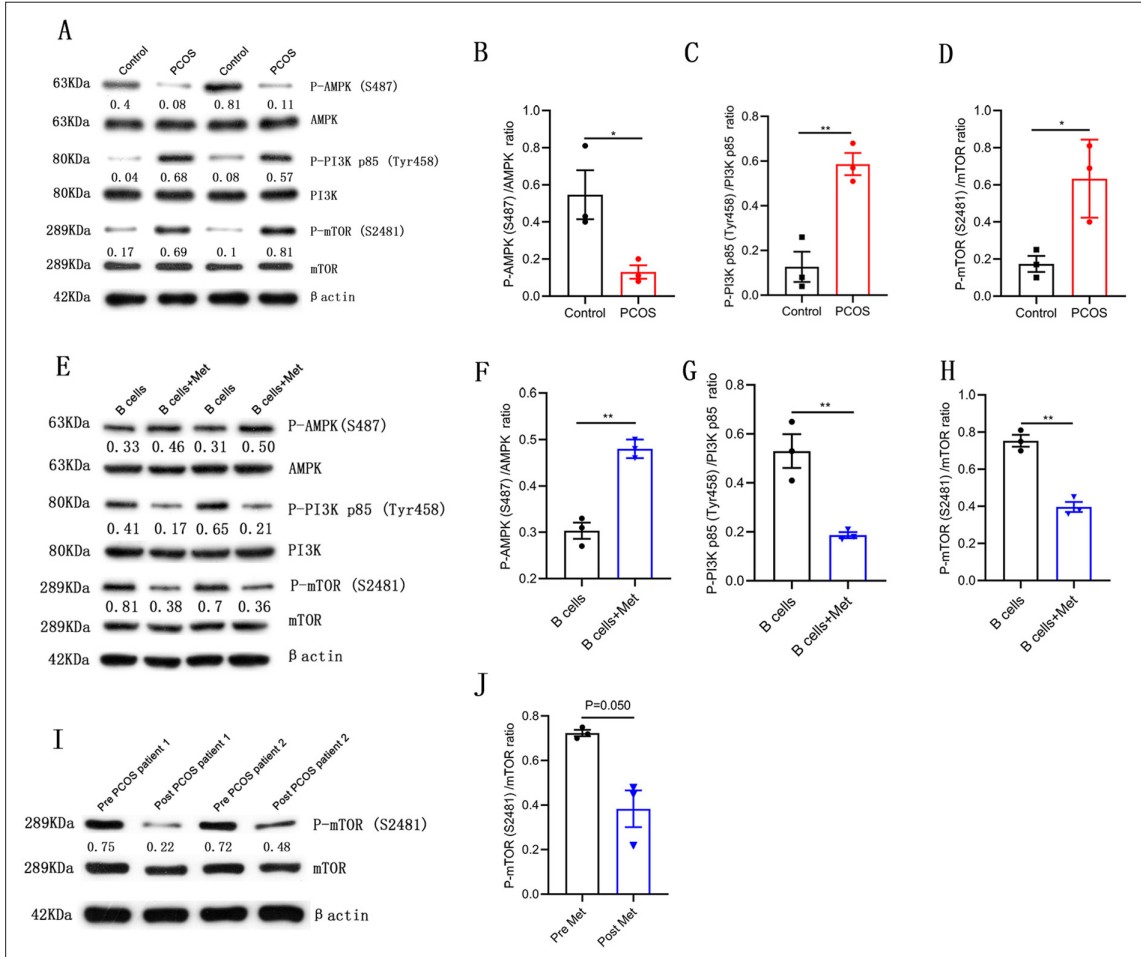

**Figure 3.** Metformin inhibits mechanistic target of rapamycin (mTOR) phosphorylation in pathological B cells. (**A–D**) Western blot of representative images and quantification analysis of the ratios of phosphorylated (P-) AMPK (S487)/AMPK, P-PI3K p85 (Tyr458)/PI3K p85, and P-mTOR (S2481)/mTOR in B cells isolated from peripheral blood (PB) of women with polycystic ovary syndrome (PCOS) (n=3 per group). (**E–H**) Western blot of representative images and quantification analysis of the ratios of P-AMPK (S487)/AMPK, P-PI3K p85 (Tyr458)/PI3K p85, and P-mTOR (S2481)/mTOR in stimulated peripheral blood B cells from women with PCOS (n=3 per group) with or without metformin (12.5 mM, Met) for 48 hr in vitro. (**I–J**) Western blot of representative images and quantification analysis of the ratio of P-mTOR (S2481)/mTOR in B cells isolated from PB of women with PCOS, before (pre) and after (post) treatment with metformin for one month (n=3 per group). p values were determined by two-tailed Student's t-test and data are presented as means ± SEM. *p<0.05; **p<0.01.

The online version of this article includes the following source data for figure 3:

**Source data 1.** Data points for graphs in *Figure 3*.

## Metformin inhibits mTOR phosphorylation in pathological B cells

Metformin is known to affect AMPK/PI3K/mTOR mediated signaling (*Chen et al., 2019*; *Lee et al., 2017*). In order to explore the molecular mechanism of metformin inhibiting TNF-α expression in B cells from women with PCOS, we analyzed the phosphorylation levels of AMPK/PI3K/mTOR signaling molecules. The results showed that the phosphorylation level of AMPK (S487) in CD19$^+$ B cells isolated from PB of women with PCOS was significantly lower but the phosphorylation levels of PI3K p85 (Tyr458) and mTOR (S2481) were significantly higher compared with control subjects (*Figure 3A–D*). In the in vitro culture system of B cells, metformin significantly increased the phosphorylation level of AMPK (S487), and inhibited the phosphorylation levels of PI3K p85 (Tyr458) and mTOR (S2481) in B cells from women with PCOS (*Figure 3E–H*). After oral metformin treatment, the phosphorylation level of mTOR (S2481) in peripheral blood B cells from women with PCOS was also significantly reduced (*Figure 3I–J*). These results suggest that the abnormal phosphorylation of mTOR (S2481) may be related to the increase of pathological TNF-α-producing B cells and metformin suppresses

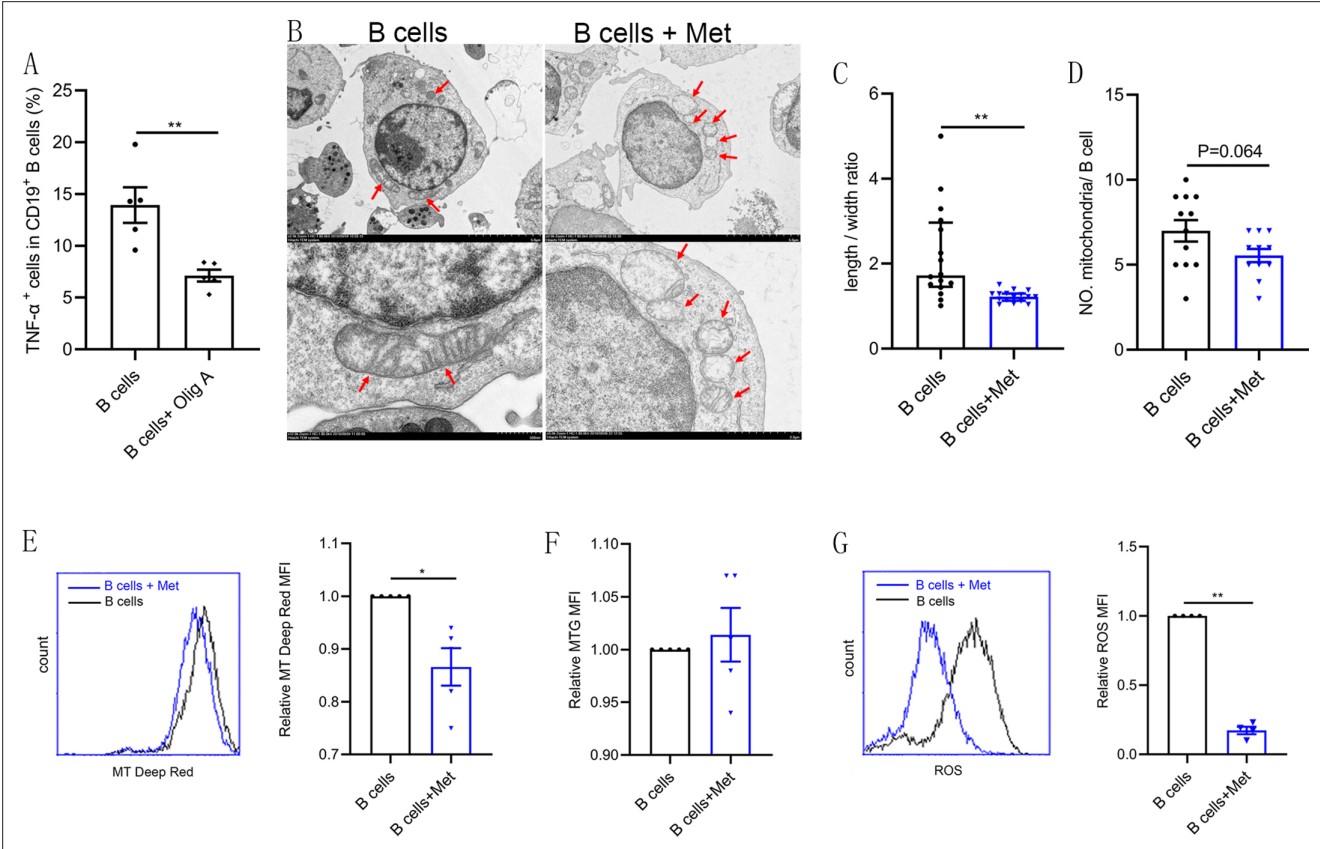

**Figure 4.** Metformin induces mitochondrial remodeling in pathological B cells. (**A**) Percentage of tumor necrosis factor-alpha plus (TNF-α$^+$) cells in CD19$^+$ B cells from women with polycystic ovary syndrome (PCOS) stimulated with B cell-targeted activator with or without oligomycin A (100 nM, Olig A) for 48 hr (n=5 per group). (B-G) B cells isolated from peripheral blood (PB) of women with PCOS were cultured in RPMI-1640 medium supplemented B cell-targeted activator with or without metformin (12.5 mM, Met) for 48 hr in vitro. (**B**) Representative transmission electron microscopy images. Up: scale bar, 5 μm; down: scale bar, 500 nm. Red arrows indicate mitochondria. (**C**) The length/width ratio of per mitochondria in B cells (n=16 per group). (**D**) The numbers of mitochondria in per B cell (n=11 per group). (**E**) Mitochondrial membrane potential (MMP) was measured using MitoTracker (MT) Deep Red by flow cytometric analysis. Shown are representative images and quantification analysis of the MT Deep Red (n=5 per group). (**F**) Mitochondrial mass was measured using MitoTracker Green (MTG) by flow cytometric analysis (n=5 per group). (**G**) ROS was measured by flow cytometric analysis. Shown are representative images and quantification analysis of the ROS level (n=4 per group). For (A) and (D), p values were determined by two-tailed Student's t-test and data are presented as means ± SEM. For (C), p values were determined by two-tailed Mann-Whitney U-test and data are presented as medians with interquartile ranges. For (E), (F), and (G), p values were determined by two-tailed paired-samples t-test and data are presented as means ± SEM. *p<0.05; **p<0.01.

The online version of this article includes the following source data for figure 4:

**Source data 1.** Data points for graphs in *Figure 4*.

TNF-α production in pathological B cells from women with PCOS through inhibiting the mTOR phosphorylation.

## Metformin induces mitochondrial remodeling in pathological B cells

Intracellular metabolism influences the production of inflammatory cytokines by immune cells (*Bharath et al., 2020*). Oligomycin A, an oxidative phosphorylation (OXPHOS) inhibitor, significantly inhibited TNF-α production in CD19+ B cells isolated from PB of women with PCOS, indicating that the level of OXPHOS in B cells affects the production of TNF-α (*Figure 4A*). Augmented tricarboxylic acid (TCA) cycle and OXPHOS activation in stimulated B cells suggests a coordinated change in mitochondrial dynamics and morphology (*Waters et al., 2018*). To evaluate the effects of metformin on mitochondria morphology and functions of B cells from women with PCOS, we first examined mitochondrial morphology by transmission electron microscopy. The results showed that the majority of mitochondria in metformin-treated B cells became round and the mitochondrial inner membrane disappeared (*Figure 4B–C*), but the mitochondrial numbers were not affected (*Figure 4D*). To further characterize the mitochondrial metabolic state, we measured the mitochondrial membrane potential (MMP), an important marker of mitochondrial activity, by staining with the mitochondrial dye MitoTracker (MT) Deep Red, and the mitochondrial mass by staining with MitoTracker Green (MTG). The results showed that metformin decreased MT Deep Red fluorescence intensity and MMP in B cells, suggesting metformin inhibited mitochondrial activity of B cells from women with PCOS (*Figure 4E*). However, metformin had no effect on mitochondrial mass (*Figure 4F*). In addition, ROS production is considered to be a biomarker of mitochondrial respiration. Excessive ROS can induce oxidative stress, ultimately causing mitochondrial dysfunction and Th17 cytokine production (*Bharath et al., 2020*). Metformin reduced ROS levels in B cells, suggesting that metformin might behave as an antioxidant to inhibit the expression of proinflammatory cytokines in pathological B cells (*Figure 4G*). These results suggest that metformin induced mitochondrial remodeling of B cells in women with PCOS, including the alteration in mitochondrial morphology, the decrease in MMP and ROS production.

## Metformin reduces glucose uptake in pathological B cells

B lymphocytes rapidly increase glucose uptake following B cell antigen receptor crosslinking (*Doughty et al., 2006*). We would like to know whether metformin affects glucose uptake of pathological B cells in women with PCOS. The 2-[N-(7-nitrobenz-2-oxa-1,3-diazol-4-yl) amino]-2-deoxy-d-glucose is a fluorescent glucose analog, which is commonly used to assess glucose uptake by cells. As shown in *Figure 5A*, the addition of metformin to the B cell culture medium significantly inhibited the uptake of 2-NBDG by B cells, suggesting that metformin affects the production of TNF-α, which may be related to the inhibition of the energy metabolism by metformin in pathological B cells. The glucose transporter is responsible for the uptake of glucose. We next detected the effects of metformin on the expression of glucose transporter. Western blot results showed that metformin significantly inhibited the expression of glucose transporter Glut 1 and Glut 4 in B cells (*Figure 5B–D*). Furthermore, we found that metformin mainly inhibited the expression of Glut 4 on B cells after a short period of incubation with B cells (only 30 min), which had no obvious effect on Glut 1 (*Figure 5E–F*). While metformin mainly inhibited the expression of Glut 1 rather than Glut 4 on B cells after 4 hr incubation (*Figure 5E–F*). It was reported that HIF1α and c-Myc promote the transcription of metabolic-related genes, including glucose transporters (*Wang et al., 2011*). We therefore detected the effect of metformin on the expression of HIF1α and c-Myc in pathological B cells from women with PCOS. The results showed that metformin significantly inhibited the expression of HIF1α and c-Myc (*Figure 5G–I*). These findings demonstrate that metformin reduces glucose uptake in pathological B cells from women with PCOS through inhibiting the expression of Glut 1 and Glut 4 and the upstream transcription factors HIF1α and c-Myc.

## Rapamycin recapitulates the effects of metformin on pathological B cells

Previous results suggest that the abnormal phosphorylation of mTOR may be related to the increase of pathological TNF-α-producing B cells and metformin inhibited mTOR phosphorylation, induced mitochondrial remodeling, inhibited TNF-α expression in pathological B cells. To further prove that mTOR is involved in these processes, we studied the effect of rapamycin, a mTOR inhibitor, on pathological

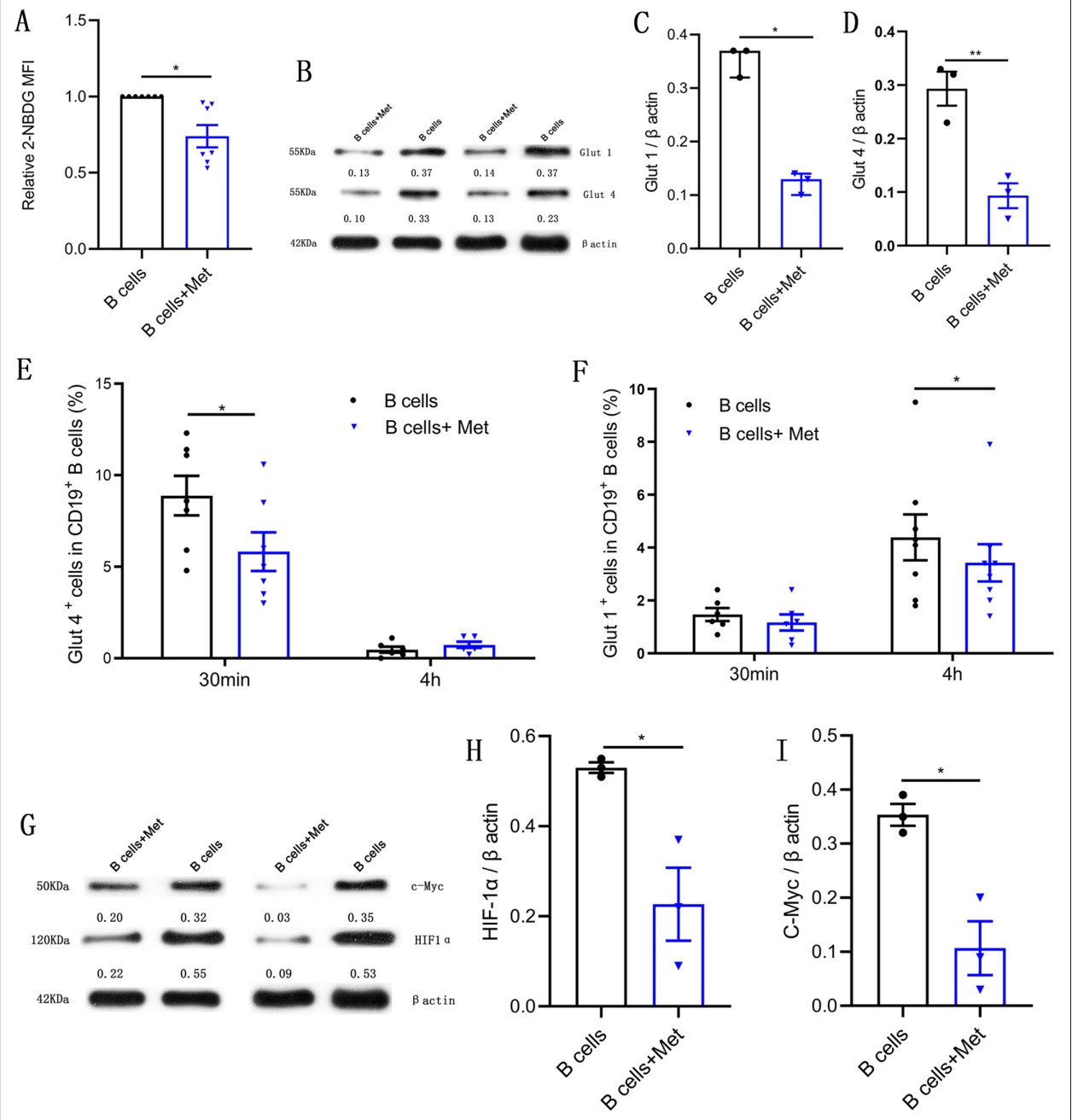

**Figure 5.** Metformin reduces glucose uptake in pathological B cells. (**A-D**) and (**G-I**) B cells isolated from peripheral blood (PB) of women with polycystic ovary syndrome (PCOS) were cultured in RPMI-1640 medium supplemented with B cell-targeted activator with or without metformin (12.5 mM, Met) for 48 hr in vitro. (**A**) Glucose uptake was measured using 2-NBDG by flow cytometric analysis (n=7 per group). (**B-D**) Glut 1 and Glut 4 expression in CD19+ B cells by Western blot. Shown are representative images and quantification analysis of ratios of Glut 1/β actin (C) and Glut 4/β actin (D) (n=3 per group). (**E-F**) Percentage of Glut 1+ cells and Glut 4+ cells in CD19+ B cells from women with PCOS were cultured in B cell culture medium with or without metformin (12.5 mM, Met) for 30 min and 4 hr (n=6–8 per group). (**G-I**) HIF1α and c-Myc expression in CD19+ B cells by Western blot. Shown are representative images and quantification analysis of ratios of HIF1α /β actin (H) and c-Myc /β actin (I) (n=3 per group). For (A), (E), (F), and (J), p values were determined by two-tailed paired-samples t-test and data are presented as means ± SEM. For (C), p values were determined by two-tailed Mann-Whitney U-test and data are presented as medians with interquartile ranges. For (D), (H), and (I), p values were determined by two-tailed Student's t-test and data are presented as means ± SEM. *p<0.05; **p<0.01.

The online version of this article includes the following source data for figure 5:

**Source data 1.** Data points for graphs in **Figure 5**.

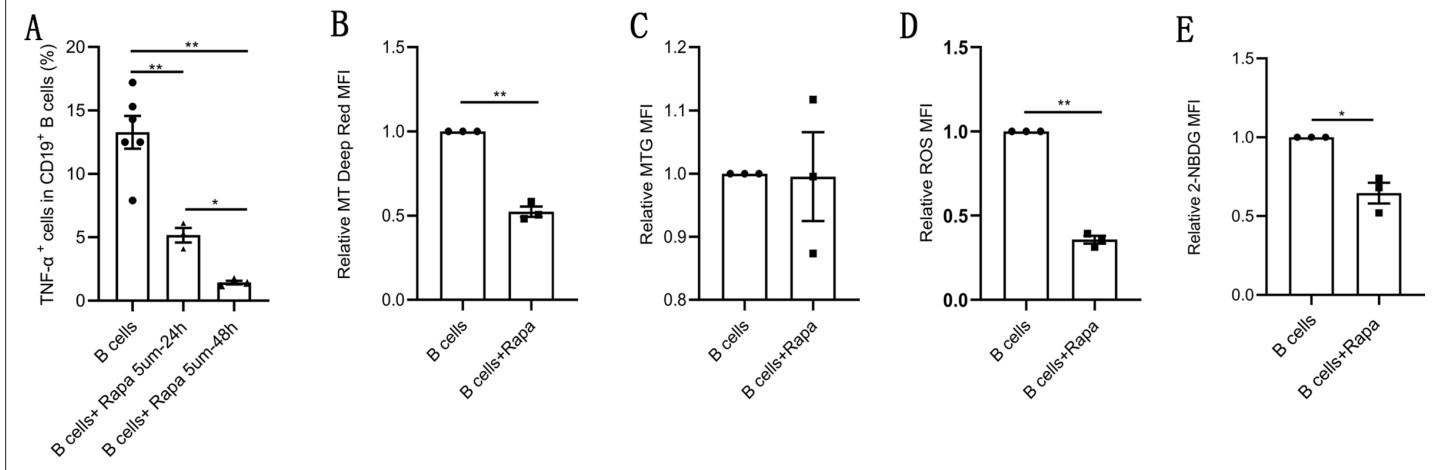

**Figure 6.** Rapamycin inhibits tumor necrosis factor-alpha (TNF-α) production, induces mitochondrial remodeling, and reduces glucose uptake in pathological B cells. (**A**) Percentage of TNF-α+ cells in CD19+ B cells from women with polycystic ovary syndrome (PCOS) stimulated with B cell-targeted activator with or without rapamycin (5 μM, Rapa) for 24 hr or 48 hr (n=3–6 per group). (**B–E**) B cells isolated from peripheral blood (PB) of women with PCOS were cultured in RPMI-1640 medium supplemented B cell-targeted activator with or without rapamycin (5μM, Rapa) for 48 hr and rapamycin was added to the culture medium on the first day (24 hr). (**B**) Mitochondrial membrane potential measured using Mito Tracker (MT) Deep Red by flow cytometric analysis (n=3 per group). (**C**) Mitochondrial mass measured using MitoTracker Green (MTG) by flow cytometric analysis (n=3 per group). (**D**) ROS measured by flow cytometric analysis (n=3 per group). (**E**) 2-NBDG measured by flow cytometric analysis (n=3 per group). For (A), p values were determined by two-tailed Student's t-test and data are presented as means ± SEM. For (B), (C), (D), and (E), p values were determined by two-tailed paired-samples t-test and data are presented as means ± SEM. *p<0.05; **p<0.01.

The online version of this article includes the following source data for figure 6:

**Source data 1.** Data points for graphs in *Figure 6*.

B cell from women with PCOS. As expected, rapamycin exhibited a time-dependent inhibitory effect on the expression of TNF-α in pathological B cells (*Figure 6A*). We also investigated the effects of rapamycin on mitochondrial metabolic state and glucose uptake, and found that rapamycin decreased the MMP, ROS levels and glucose uptake in pathological B cells (*Figure 6B–E*). These results were consistent with the effect of metformin on pathological B cells, demonstrating that mTOR phosphorylation was involved in the inhibitory effect of metformin on the production of TNF-α and metabolic reprogramming of B cells in women with PCOS.

## Metformin improves DHEA-induced PCOS phenotypes

To confirm the role of metformin in PCOS, we investigated the therapeutic effect of metformin on the PCOS mouse model induced by DHEA and evaluated the effect of metformin on ovarian structure and function, glucose metabolism as well as systemic inflammation. *Figure 7A* showed the schematic diagram of the experimental protocols for metformin in the treatment of DHEA-induced PCOS mice. DHEA-induced PCOS mice had fewer corpora lutea, more cystic follicles, and thicker theca cell layers, which were consistent with the changes in ovarian morphology in women with PCOS (*Figure 7B–D*). In addition, estrous cycling was disrupted in PCOS mice as determined by vaginal cytology (*Figure 7E*). After metformin treatment, the PCOS mice showed an increased number of corpora lutea and normal estrous cycles (*Figure 7B–E*). These results suggest that metformin treatment can reverse ovulatory dysfunction in DHEA-induced PCOS mice. In the glucose tolerance tests (GTTs), PCOS mice showed impaired glucose tolerance, while metformin could markedly lower glucose levels and GTT area under the curve (GTT AUC), indicating that metformin can restore the impaired glucose tolerance of PCOS mice (*Figure 7F–G*). We further detected the level of TNF-α in serum, the results showed that the level of TNF-α was higher in mice with PCOS compared with control subjects, and metformin inhibited this increase (*Figure 7H*). These data suggest that metformin treatment reverses abnormal ovarian morphology, disturbed estrous cycle, impaired glucose metabolism and abnormal serum TNF-α levels in DHEA-induced PCOS mice.

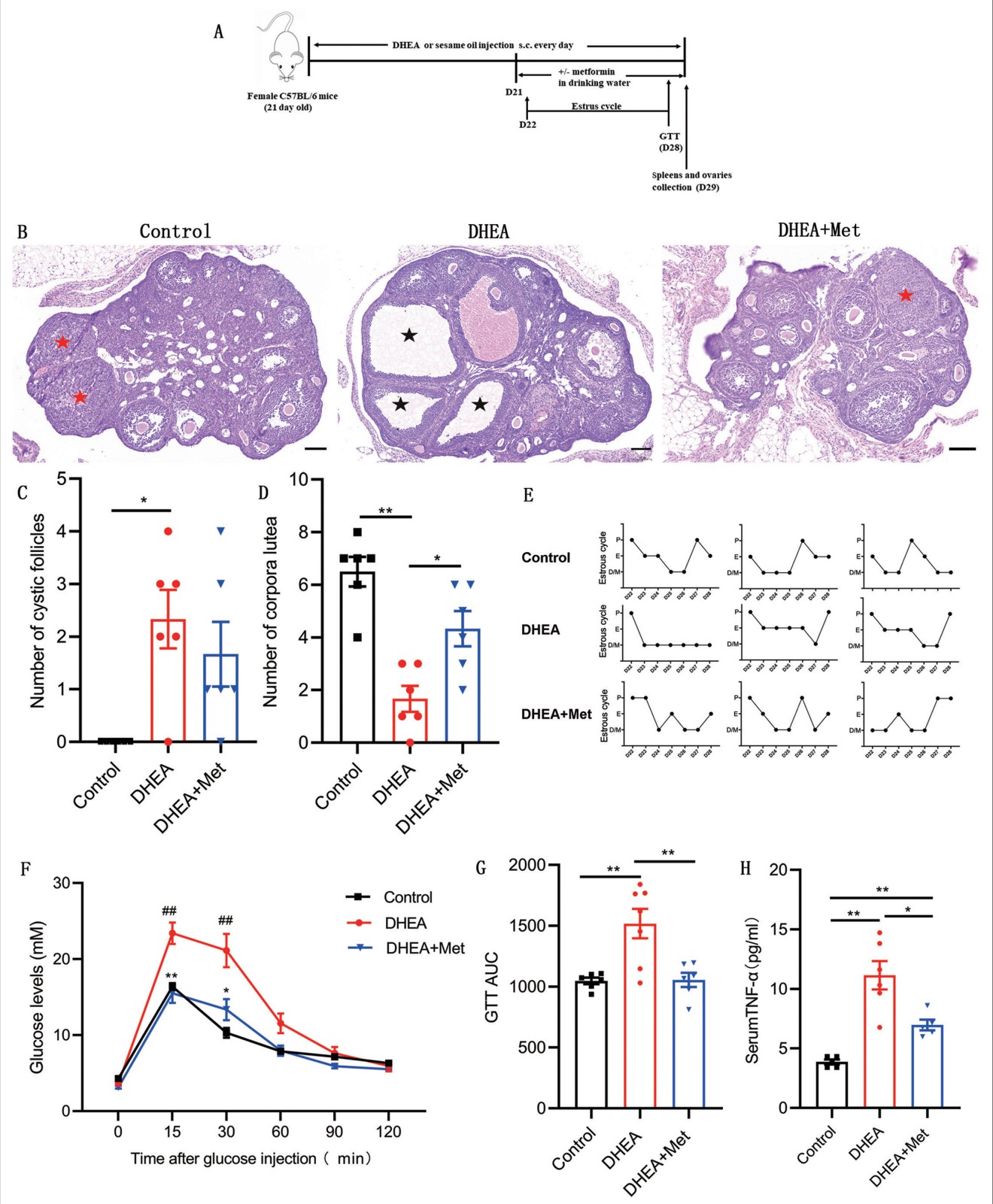

**Figure 7.** Metformin treatment improves the polycystic ovary syndrome (PCOS) pathological phenotypes in dehydroepiandrosterone (DHEA)-induced mice. (**A**) Schematic diagram of the experimental protocols for metformin in the treatment of DHEA-induced PCOS mice. (**B**) Representative ovary sections with Hematoxylin and eosin staining. Black asterisks indicate cystic follicles, and red asterisks indicate corpora lutea. Scale bar: 100 µm. (**C**) Quantitative analysis of cystic follicles (n=4–6 mice per group). (**D**) Quantitative analysis of corpora lutea (n=4–6 mice per group). (**E**) Representative

*Figure 7 continued on next page*

*Figure 7 continued*

estrous cycles. (**F**) Glucose tolerance test (GTT) (n=4–6 mice per group). (**G**) GTT area under the curve (GTT AUC) (n=4–6 mice per group). (**H**) Serum tumor necrosis factor-alpha (TNF-α) concentrations were assessed with multiplex assays (n=4–6 mice per group). p values were determined by one-way ANOVA with Bonferroni's multiple comparison post-hoc test and data are presented as means ± SEM. *p<0.05; **p<0.01. For F, *p<0.05; **p<0.01 DHEA + metformin (Met) versus DHEA. #p<0.05; ##p<0.01 DHEA versus the control.

The online version of this article includes the following source data for figure 7:

**Source data 1.** Data points for graphs in *Figure 7*.

## Effect of metformin on B cells from DHEA-induced PCOS mice

We further evaluated the effects of metformin on the production of TNF-α, molecular mechanism, and metabolism in B cells isolated from spleen of DHEA-induced PCOS mice. The percentage of TNF-α$^+$ cells in CD19$^+$ B cells was notably higher in mice with PCOS than that in control groups and metformin inhibited the expression of TNF-α in pathological B cells (*Figure 8A*). Simultaneously, metformin significantly increased the phosphorylation level of AMPK (S487), and inhibited the phosphorylation levels of PI3K p85 (Tyr458) and mTOR (S2481) in splenic B cells from mice with PCOS (*Figure 8B*). These results were consistent with the results in women with PCOS. We next examined cellular metabolites by mass spectrometry. Metformin increased the accumulation of glycolytic lactate and TCA cycle intermediates (succinate, fumarate, and oxaloacetate) in splenic B cells (*Figure 8C–D*). These data demonstrate that metformin can inhibit TNF-α production, affect AMPK/PI3K/mTOR signal pathway, and induce metabolic reprogramming in pathological B cells from DHEA-induced PCOS mice.

## Discussion

Chronic inflammation is a typical characteristic of PCOS, in which TNF-α plays an important role. During the periovulatory stage, injection of TNF-α into the rat bursa reduced the number of released oocytes and induced GCs apoptosis and autophagy, suggesting that TNF-α is an important mediator of ovulation (*Yamamoto et al., 2015*). Etanercept, a TNF-α inhibitor, could alleviate PCOS phenotypes, which further demonstrates the effect of TNF-α on the development of PCOS (*Lang et al., 2019*). In women with PCOS, the increase of TNF-α levels was positively correlated with biochemical parameters: DHEA and glucose levels (*Szczuko et al., 2018*). In our study, women with PCOS and DHEA-induced PCOS mice both exhibited elevation in serum levels of TNF-α, which were consistent with previous reports (*Bhatnager et al., 2019*; *Spritzer et al., 2015*; *Szczuko et al., 2018*); TNF-α significantly increased the expression of IL-6, IL-8, CCL2, and CCL20 in GCs, leading to inflammation of GCs, which may eventually lead to ovulation disorders in PCOS.

Our previous results reported that mice with CD19$^+$ B cell depletion failed to develop PCOS, indicating that CD19$^+$ B cells play a critical role in the etiology of PCOS (*Xiao et al., 2019*). B cells play a role in chronic inflammatory and autoimmune diseases by secreting proinflammatory cytokines (*Shen and Fillatreau, 2015*). Interestingly, we found that peripheral blood B cells from women with PCOS and B cells from DHEA-induced PCOS mice both produced higher levels of TNF-α, and the percentage of TNF-α$^+$ cells in human peripheral CD19$^+$ B cells was positively associated with the serum AMH levels. It is reported that TNF-α-deficient splenic B cells were less efficient than wild type B cells to amplify IFN-γ expression by T cells during the course of the Th1 inflammatory response to *Toxoplasma gondii* infection (*Menard et al., 2007*). Atherosclerosis is reduced in mice with TNF-α-deficient B cells; B cells-derived TNF-α is a key cytokine that promotes atherosclerosis development through augmenting macrophage TNF-α production to induce cell death and inflammation that promote plaque vulnerability (*Tay et al., 2016*). These findings suggest that pathological TNF-α-producing B cells may contribute to the development of PCOS by promoting inflammatory response.

In this study, we propose a novel mechanism that metformin alleviates PCOS by inhibiting TNF-α production in pathological B cells. In a mouse model of SLE, metformin suppresses systemic autoimmunity through inhibiting B cell differentiation into autoreactive plasma cells, as well as germinal center formation (*Lee et al., 2017*). Metformin attenuates bleomycin-induced scleroderma by reducing splenic germinal center formation (*Wang et al., 2019*). Our studies showed that metformin was able to reduce the expression of TNF-α in B cells from women with PCOS in vitro and in vivo. Administration of metformin improved DHEA-induced PCOS phenotypes and also inhibited TNF-α expression in

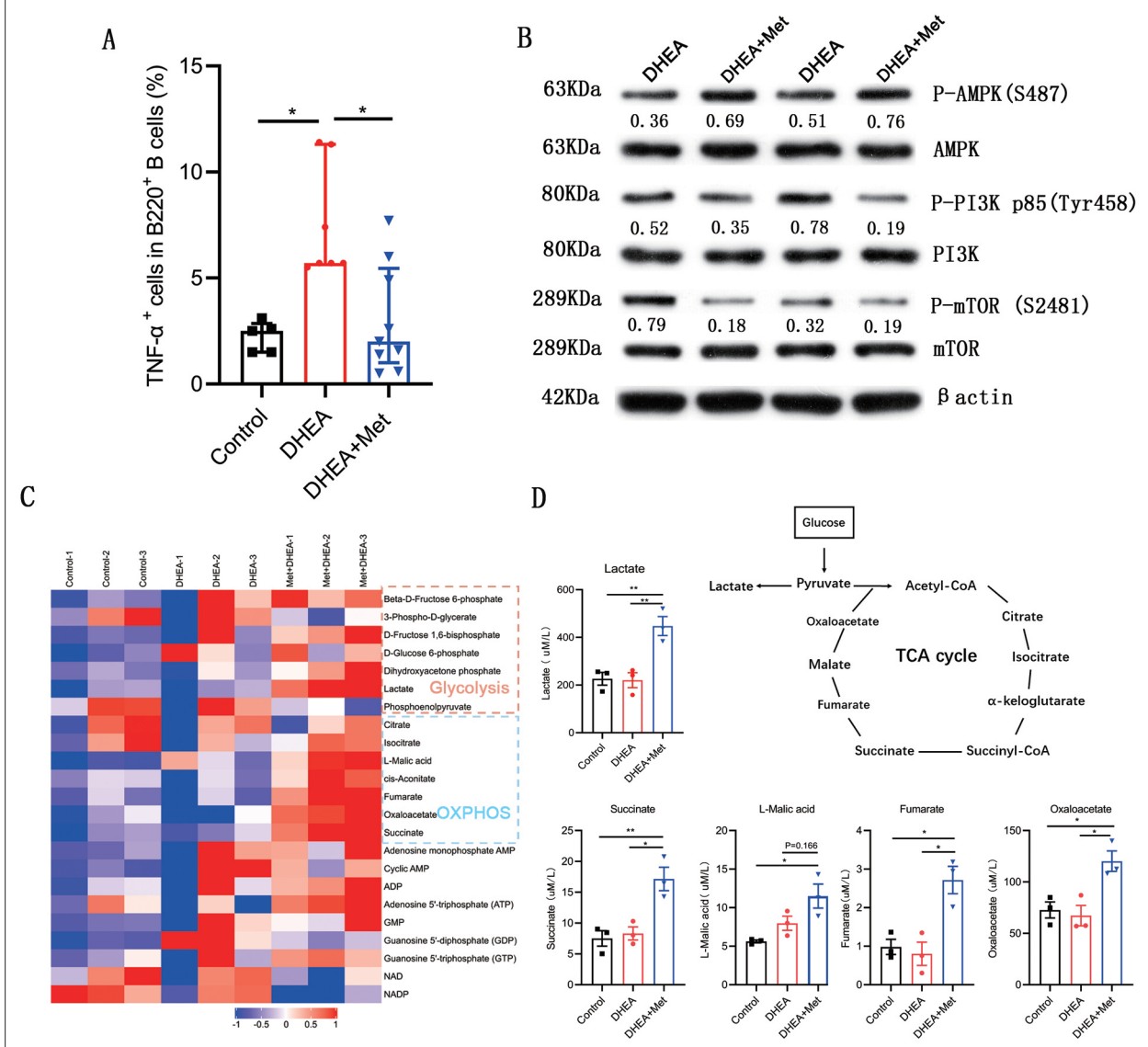

**Figure 8.** Metformin inhibits tumor necrosis factor-alpha (TNF-α) production, alters AMPK/PI3K/mTOR phosphorylation and induces metabolic reprogramming in pathological B cells from dehydroepiandrosterone (DHEA)-induced polycystic ovary syndrome (PCOS) mice. (**A**) Percentage of TNF-α⁺ cells in splenic CD19⁺ B cells by flow cytometric analysis (n=5–9 mice per group). (**B**) Western blot of representative images and quantification analysis of the ratios of phosphorylated (P-) AMPK (S487)/AMPK, P-PI3K p85 (Tyr458)/PI3K p85, and P-mTOR (S2481)/mTOR in splenic B cells from DHEA-induced PCOS mice (n=2 mice per group). (**C**) The heat-map analysis of energy metabolites in splenic B cells was analyzed by mass spectrometry (n=3 mice per group). (**D**) Differential energy metabolites were quantitatively analyzed by mass spectrometry (n=3 mice per group). For (A), p values were determined by Kruskal-Wallis test followed by Dunn's post-hoc test and data are presented as medians with interquartile ranges. For (D), p values were determined by one-way ANOVA with Bonferroni's multiple comparison post-hoc test and data are presented as means ± SEM. *p<0.05; **p<0.01.

The online version of this article includes the following source data for figure 8:

**Source data 1.** Data points for graphs in *Figure 8*.

splenic B cells. Thus, we speculate that the therapeutic effect of metformin on PCOS may be related to the inhibition of TNF-α production in pathological B cells.

Mitochondria is the main subcellular target of metformin, which inhibits mitochondrial respiratory chain activity (*Rena et al., 2017*). In our study, metformin induced mitochondrial remodeling in B cells from women with PCOS. The destruction of mitochondrial inner membrane and the decrease of MMP and ROS production were related to the inhibition of mitochondrial respiratory chain activity by metformin. Reduced ROS levels -are associated with increased aerobic glycolysis (*Beltran et al.,*

*2020*). Metformin has been reported to induce glycolysis in rat brain astrocytes (*Westhaus et al., 2017*). We also found that metformin increased glycolytic lactate production in splenic B cells from DHEA-induced PCOS mice. Simultaneously, metformin also increased TCA cycle intermediates accumulation in splenic B cells. In addition, we found that metformin reduced glucose uptake in pathological B cells from women with PCOS through inhibiting the expression of Glut 1 and Glut 4, and the upstream transcription factors HIF1α and c-Myc. These results were consistent with the effect of metformin on T cells. Metformin-treated antigen receptor activated T cells showed reduced glucose uptake compared to control antigen-activated cells, and metformin treatment blocked T cell antigen receptor-induced expression of c-Myc, HIF-1α, and Glut 1 in T cells (*Zarrouk et al., 2014*). We also found that OXPHOS inhibitor oligomycin A inhibited TNF-α production in B cells, indicating that intracellular metabolism affects the production of TNF-α in B cells. These findings suggest that metformin inhibits the production of TNF-α in pathological B cells, which is related to the regulation of metformin on B cell energy metabolism. mTOR activity plays a critical role in B cell function. Conditional deletion of the mTOR in B cells impairs germinal center differentiation and high-affinity antibody responses (*Zhang et al., 2013*).

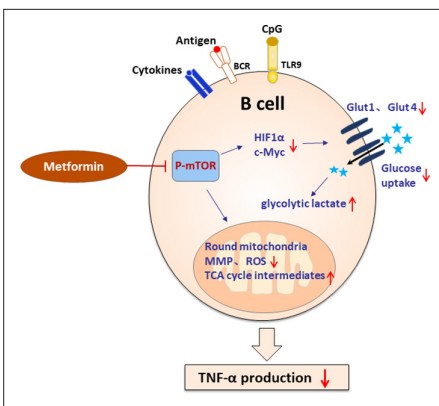

**Figure 9.** Schematic diagram of the effect of metformin on tumor necrosis factor-alpha (TNF-α)-producing B cells in polycystic ovary syndrome (PCOS). In peripheral blood B cells from women with PCOS, metformin inhibits mechanistic target of rapamycin (mTOR) phosphorylation; reduces glucose uptake by downregulation of Glut 1 and Glut 4 expression, and the upstream transcription factors HIF1α and c-Myc; alters the mitochondrial morphology; decreases the mitochondrial membrane potential (MMP) and ROS levels. In splenic B cells from dehydroepiandrosterone (DHEA)-induced PCOS mice, metformin inhibits mTOR phosphorylation; stimulates glycolytic lactate and TCA cycle intermediates accumulation. These alterations ultimately result in the decrease of TNF-α production by pathological B cells in PCOS.

Suppressing mTOR activity by rapamycin markedly inhibits B cell activation and antibody secretion (*Sintes et al., 2017*). Our results showed that the phosphorylation level of mTOR in CD19[+] B cells from women with PCOS was significantly higher compared with control subjects; and rapamycin markedly inhibited TNF-α expression in B cells from women with PCOS. These results suggest that mTOR phosphorylation is involved in the regulation of TNF-α expression in pathological B cells. The mTOR is the cellular target of metformin. Inhibition of B cell differentiation and germinal center formation in a mouse model of SLE by metformin is characterized by decreased mTOR phosphorylation in B cells (*Lee et al., 2017*). Metformin exerts the inhibitory effects on human soluble B-cell activating factor-induced B-cell proliferation/viability through hampering the mTOR pathway (*Chen et al., 2021b*). In our study, the mTOR phosphorylation in B cells was significantly reduced after oral administration of metformin in PCOS patients and mouse model. In in vitro experiments, metformin showed a similar inhibitory effect on mTOR phosphorylation in B cells. The inhibitory effect of metformin on mTOR phosphorylation may be contributed to its suppression on TNF-α expression in pathological B cells. In addition, the mTOR pathway regulates various aspects of metabolism including mitochondrial function and glucose metabolism (*Bjedov and Rallis, 2020*; *Chen et al., 2021a*). Rapamycin reduces mitochondrial biogenesis, mitochondrial oxygen consumption, and intracellular $H_2O_2$ production in human mesenchymal stem cells during adipocyte differentiation (*Tormos et al., 2011*). Rapamycin reduces glucose uptake and the expression of glycolytic enzymes in T cells during Treg polarization (*Chen et al., 2021a*). Our results showed that rapamycin decreased the MMP, ROS levels, and glucose uptake in pathological B cells from women with PCOS. These data suggest that metformin induces metabolic reprogramming of pathological B cells through mTOR pathway. We will further investigate the regulation mechanism of metformin on mTOR signaling in TNF-α-producing pathological B cells.

In summary, we found that pathological TNF-α-producing B cells are involved in the pathological process of PCOS. Metformin inhibits mTOR phosphorylation and affects metabolic reprogramming, and then inhibits TNF-α expression in pathological B cells (*Figure 9*). These data presented here

may provide new insights into the mechanism of metformin in the treatment of PCOS and specific targeting TNF-α-producing B cells may be a potential therapy.

## Materials and methods

### Mice

All animal experiments were approved by the ethic review committee of Central South University (NO.2019-S111). PCOS in mice was induced with DHEA (Sigma-Aldrich) as previously described (*Xiao et al., 2019*). Briefly, female prepuberal (21-day-old) C57BL/6 mice received daily subcutaneous injections of DHEA (6 mg/100 g body weight, dissolved in 0.1 ml sesame oil [Acros]) for 21 days. After PCOS is established, PCOS mice were randomly divided into two groups. PCOS mice continued to be injected with DHEA and were treated with or without metformin in drinking water (200 mg kg$^{-1}$ per day) for another 8 days. The control group received corresponding daily injections of 0.1 ml sesame oil. Estrous cycle stages were determined by light microscope analysis of vaginal epithelial cell smears. After 29 days of treatments, all mice were anesthetized and killed, and their blood, spleens and ovaries were harvested. The spleens of mice were mechanically disrupted and filtered through 70 µm cell strainers to obtain single-cell suspensions. Splenic B cells were isolated with the CD45R (B220) cell isolation kit (Miltenyi Biotec). The purity of B220 cells was >90%. Ovaries were fixed in a 4% formaldehyde solution. The ovaries were serially sectioned at 5 µm, and every tenth section was mounted on a glass slide and observed under light microscopy for histomorphology examinations. The number of corpora lutea and cystic follicles of the ovaries werecounted.

### Glucose tolerance test

For GTT experiment, mice were fasted for 16 hr and intraperitoneally injected with glucose solution (Dashi, China; 2 g/kg body weight). Blood glucose levels were measured from tail blood using glucose meters (Sannuo) at 0, 15, 30, 60, 90, and 120 min after glucose injection.

### Human subjects

Patients and control subjects were recruited from the Reproductive & Genetic Hospital of CITIC-Xiangya. This study was approved by the ethic committee of the Reproductive & Genetic Hospital of CITIC-Xiangya (NO.LL-SC-2015–007), and all participants provided informed consent. Women with PCOS were diagnosed according to the 2003 Rotterdam criteria with at least two of the following features: oligo- or anovulation, clinical and/or biochemical hyperandrogenism, and polycystic ovaries identified by ultrasound (≥12 small follicles with diameters of 2–9 mm in at least one ovary and/or an ovary volume >10 ml). Patients with any other clinical conditions were excluded, such as non-classic 21-hydroxylase deficiency, congenital adrenal hyperplasia, thyroid dysfunction, Cushing's syndrome, or significant elevations in serum prolactin ('Revised 2003 consensus on diagnostic criteria and long-term health risks related to polycystic ovary syndrome (PCOS),' 2004). Healthy female control subjects were required to have (i) regular menstrual cycles (menstrual periods of 26–35 days) and no clinical manifestations of PCOS, (ii) normal results from sex hormone and endocrine tests (for thyroid dysfunction or diabetes), (iii) no ultrasonographic evidence of polycystic ovaries, (iv) no evidence of systemic or inflammatory diseases, and (v) not received hormonal therapy (including oral contraceptives) or drug therapy in the last 3 months. Women with PCOS were taking 500 mg of metformin (oral tablet), two times per day for 1 month. Heparinized PB samples were collected from women with PCOS and healthy subjects. AMH assay was performed using the Uranus AE automatic enzyme immunoassay (Aikang Medtech).

### Cell culture

Human PBMCs were isolated by Ficoll–Hypaque density gradient centrifugation, followed by negative selection with the B cell isolation kit (Invitrogen). The purity of CD19$^+$ B cells was >90%. B cells were cultured in RPMI-1640 containing 10% fetal bovine serum (FBS, Natocor), 2.5 µg/ml F(ab')$_2$ fragment goat anti-human IgA + IgG + IgM (Jackson ImmunoResearch), 100 ng/ml CD40 L (Gibco), 10 ng/ml IL-4 (Gibco), 10 ng/ml IL-10 (Gibco), 2.5 µg/ml CpG oligodeoxynucleotide 2006 (TCGTCGTTTTGT CGTTTTGTCGTT; Sangon Biotech), and 50 U/ml IL-2 (Gibco) for 2 days. Metformin (12.5 mM, Sigma), or oligomycin A (100 nM, MCE) was added to the culture medium, respectively. Rapamycin (5 µM, LC

Labs) was added to the culture medium on the first day (24 hr) or 48 hr, respectively. The supernatants and cells were collected for analysis.

GCs were retrieved from follicular fluid of 7 healthy female subjects undergoing In Vitro Fertilization (IVF). GCs were cultured in RPMI-1640 (Gibco) supplemented with 10% FBS (Natocor), and with or without 10 ng/ml TNF-α (R&D) for 2 days. Cultured cells were lysed in TriPure isolation reagent (Roche), with storage at –80°C until RNA extraction.

## Flow cytometry

Cell surface staining was performed with human CD19-PE (clone HIB 19; BioLegend) antibodies, Glut 4-Alexa Fluor 488 (clone 925932, R&D) and Glut 1-APC (clone 202915, R&D). For intracellular TNF-α staining, stimulated B cells were cultured in the presence of phorbol 12-myristate 13-acetate (PMA, 50 ng/ml; Sigma-Aldrich), ionomycin (1 µg/ml; Sigma-Aldrich), and monensin (1: 1000 dilutions, BioLegend) for 5 hr. Cell surface staining was performed with mouse B220-PerCP/Cy5.5 (clone RA3-6B2; BioLegend) or human CD19-PE (clone HIB 19; BioLegend) antibodies, and then cells were fixed, permeabilized with Cytofix/Cytoperm and Perm/Wash solutions (Becton Dickinson), and subsequently stained with mouse TNF-α-APC (clone MP6-XT22; BioLegend) or human TNF-α- PerCP (clone Mab11; BioLegend) antibodies.

To measure glucose uptake, cells were incubated with 10 µM 2-NBDG (Cayman) for 30 min at 37°C in complete media. For mitochondrial assays, the total mitochondrial mass and membrane potential were detected using Mito Tracker Green (50 nM, Invitrogen) and Mito Tracker Deep Red dyes (100 nM, Invitrogen), respectively, for 30 min at 37°C. For total cellular ROS staining, cells were incubated with ROS assay stain (Invitrogen) for 60 min at 37°C in culture media. For cell apoptosis staining, cells were stained with Annexin V and 7-AAD solution diluted with Annexin Binding Buffer (Becton Dickinson) for 30 min at room temperature. Cell proliferation was measured by loading of cells with 1 µM CFSE (eBioscience). All measurements described above including intracellular cytokine levels, cell proliferation, apoptosis, mitochondrial mass and membrane potential, and ROS production as well as 2-NBDG uptake were conducted using a fluorescence-activated cell sorting (FACS) Aria I cytometer (Becton Dickinson) and analyzed using FACS Diva software (BD Biosciences).

## Quantitative PCR

Total RNA was extracted with TriPure Isolation Reagent (Roche), and cDNA was synthesized with the GoScript Reverse Transcription System (Promega). Quantitative PCR was performed using LightCycler 480 SYBR Green I Master (Roche) using the following gene specific primers: GADPH:GTCAAG GCTGAGAACGGGAA (forward) and TCGCCCCACTTGATTTTGGA (reverse), TNFA:TCCTTCAGACA CCCTCAACC (forward) and AGGCCCCAGTTTGAATTCTT (reverse), *CCL2*:CAGCCAGATGCAATCA ATGCC (forward) and TGGAATCCTGAACCCACTTCT (reverse), *CCL20*:TGCTGTACCAAGAGTT TGCTC (forward) and CGCACACAGACAACTTTTTCTTT (reverse), IFNG:TCGGTAACTGACTTGA ATGTCCA (forward) and TCGCTTCCCTGTTTTAGCTGC (reverse), *IL6*:ACTCACCTCTTCAGAACGAA TTG (forward) and CCATCTTTGGAAGGTTCAGGTTG (reverse), *IL8*:TTTTGCCAAGGAGTGCTAAAGA (forward) and AACCCTCTGCACCCAGTTTTC (reverse). mRNA expression was normalized to that of the housekeeping gene glyceraldehyde-3-phosphate dehydrogenase using the $2^{-\Delta CT}$ method. Relative mRNA levels were calculated by normalizing gene expression to the controls that was set to 1.

## Western blot analysis

Cells were solubilized in buffer RIPA lysis buffer (Applygen). Protein concentrations were determined by using a bicinchoninic acid assay kit (Pierce Diagnostics). Proteins were separated by 10% (wt/vol) SDS-PAGE and transferred to polyvinylidene difluoride membranes (Millipore) that were then blocked in 5% (wt/vol) skim milk in Tris-buffered saline with Tween 20 and incubated with the following primary antibodies overnight at 4°C: anti-human TNF-α (1:300, 60291–1-Ig, Proteintech), anti-PI3 kinase p85 (1:200, bs-0128r, Bioss), anti-phospho-PI3 kinase p85 (Tyr458)/p55 (Tyr199) (1:1000, 4228 s, CST), anti-AMPK (1:300, 10929–2-AP, Proteintech), anti-phospho-AMPKα1 (Ser473) (1:800, ab131357, Abcam), anti-mTOR (1:500, 20657–1-AP; Proteintech), anti-phospho-mTOR (S2481) (1:1000, ab137133, Abcam), anti-c-Myc (1:2000, 10828–1-AP, Proteintech), anti-HIF-1α (5 µg/ml, ab1, Abcam), anti-Glut 1 (1:1000, ab652, Abcam), anti-Glut 4 (1µg/ml, ab33780, Abcam), and anti-β-actin (1:5000, 60008–1-Ig, Proteintech). The membranes were then incubated with horseradish peroxidase-conjugated

secondary antibodies. Detection occurred using chemiluminescent visualization. Quantitative densitometric analysis of the immunoblotted bands was performed using Quantity One (Bio-Rad).

## Multiplex and enzyme-linked immunosorbent assays

Serum samples were collected and stored at –80°C until analysis. Milliplex map kits (Millipore) were used to quantitate serum levels of human and mouse serum cytokines using a Luminex 200 (Luminex) according to the manufacturer's instructions. TNF-α production was determined in supernatants using commercial enzyme-linked immunosorbent assay kits (R&D) following the instructions of the manufacturer.

## Transmission electron microscopy

Cells were spun down and fixed in transmission electron microscopy (TEM) fixative (Servicebio) for 2 hr. Fixed cells were processed in a standard manner and embedded in EMbed 812 (SPI). Ultrathin sections (60 nm) were cut, mounted on copper grids, and stained with uranyl acetate and lead citrate using standard methods. Stained grids were examined and photographed using a HT7800 transmission electron microscope (HITACHI). The length and width of mitochondria were calculated using ImageJ software (NIH).

## Metabolomics and metabolite quantification

To analyze metabolomics and metabolite quantification, mouse splenic B cells ($10^7$) were isolated andanalyzed at Applied Protein Technology (APT, Shanghai). Cells were processed in a standard manner. Analyses were performed using an ultra-performance liquid chromatography system (1290 Infinity LC, Agilent Technologies) and a QTRAP mass spectrometer (AB Sciex 5500).

## Statistical analysis

Statistical analyses were performed using SPSS 22.0 and GraphPad Prism 8 statistical software. The sample distribution was determined by the Shapiro-Wilk normality test. The statistical significance of differences between two groups was determined using two-tailed unpaired Student's t-tests or paired-samples t test. One-way ANOVA followed by Bonferroni's multiple comparison post-hoc test was used to evaluate the statistical significance of differences among three groups. For the non-parametric tests, the two-tailed Mann-Whitney U-test was used to evaluate statistical significance between two groups. The Kruskal-Wallis test was used to analyze differences among three experimental groups, followed by Dunn's post-hoc analysis. Data are shown as means ± SEM or as medians with interquartile ranges. Correlations were examined by Pearson's correlation tests. $*p<0.05$, $**p<0.01$, NS = not significant.

# Acknowledgements

The authors thank the patients and staff at the Reproductive and Genetic Hospital of CITIC-Xiangya for their assistance. Funding: this work was supported by the National Natural Science Foundation of China (81971355), the China Postdoctoral Science Foundation Funded Project (2020M672494) and the Hunan Provincial Natural Science Foundation of China (2021JJ40184).

# Additional information

### Competing interests

Na Xiao, Jie Wang, Xian Su, Jing Peng, Chao Yang, Ge Lin, Guangxiu Lu, Lamei Cheng: Employee of Hunan Guangxiu Hi-tech Life Technology Co, Ltd. The other authors declare that no competing interests exist.

## Funding

| Funder | Grant reference number | Author |
|---|---|---|
| National Natural Science Foundation of China | 81971355 | Lamei Cheng |
| China Postdoctoral Science Foundation | 2020M672494 | Na Xiao |
| National Science Foundation of China | 2021JJ40184 | Na Xiao |

The funders had no role in study design, data collection and interpretation, or the decision to submit the work for publication.

## Author contributions

Na Xiao, Conceptualization, Data curation, Formal analysis, Investigation, Project administration, Resources, Supervision, Validation, Writing – original draft, Writing – review and editing; Jie Wang, Data curation, Formal analysis, Investigation, Validation; Ting Wang, Xingliang Xiong, Junyi Zhou, Xian Su, Jing Peng, Chao Yang, Investigation; Xiaofeng Li, Resources; Ge Lin, Guangxiu Lu, Fei Gong, Conceptualization, Resources; Lamei Cheng, Conceptualization, Data curation, Formal analysis, Project administration, Resources, Supervision, Validation, Writing – review and editing

## Author ORCIDs

Na Xiao ⓘ http://orcid.org/0000-0003-2376-4073
Lamei Cheng ⓘ http://orcid.org/0000-0002-8685-3674

## Ethics

Patients and control subjects were recruited from the Reproductive & Genetic Hospital of CITIC-Xiangya. This study was approved by the ethic committee of the Reproductive & Genetic Hospital of CITIC-Xiangya (NO.LL-SC-2015-007), and all participants provided informed consent.
All animal experiments were approved by the ethic review committee of Central South University (NO.2019-S111).

## Decision letter and Author response

Decision letter https://doi.org/10.7554/eLife.74713.sa1
Author response https://doi.org/10.7554/eLife.74713.sa2

---

# Additional files

## Supplementary files

• Transparent reporting form

## Data availability

All data generated or analysed during this study are included in the manuscript and supporting file; Source Data files have been provided for Figures 1, 2, 3, 4, 5, 6, 7 and 8. Figure 1-source data 1, Figure 2-source data 1, Figure 3-source data 1, Figure 4-source data 1, Figure 5-source data 1, Figure 6-source data 1, Figure 7-source data 1, Figure 8-source data 1 contain the numerical data used to generate the figures.

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
