## [Editor Report]

This study confirms that TNF-α is increased in peripheral blood B cells from polycystic ovary syndrome (PCOS) and metformin decreases production. The study further suggests potential mechanisms for the increase in TNF-α and reduction due to metformin. This is demonstrated in humans as well as in a mouse model of PCOS. Overall, this is a well-designed study demonstrating the impact of metformin on immune function in PCOS.

---

## [Decision Letter]

**Decision letter after peer review:**

Thank you for submitting your article "Metformin abrogates pathological TNF-α-producing B cells through mTOR-dependent metabolic reprogramming in polycystic ovary syndrome" for consideration by *eLife*. Your article has been reviewed by 2 peer reviewers, and the evaluation has been overseen by a Reviewing Editor and Ricardo Azziz as the Senior Editor. The reviewers have opted to remain anonymous.

The reviewers have discussed their reviews with one another, and the Reviewing Editor has drafted this to help you prepare a revised submission. Please respond to the Recommendations to Authors below.

*Reviewer #1 (Recommendations for the authors):*

Better final culmination of data possible final diagram/schematic of pathway leading to TNF-α elevations in PCOS B cells, downstream effects and impact of Metformin.

*Reviewer #2 (Recommendations for the authors):*

The authors state the role of TNF-α in PCOS is mainly driven by mTOR pathway, validated that level of TNF-α is contributed mainly by Bcells. Authors further showed the molecular mechanism involved in Bcells when exposed to metformin drug. Overall, the study is well written, and the finding is important for a logical design of immunotherapeutic strategy for PCOS. Following comments are to improve the manuscript.

General Issues:

1) Kindly keep same notion as TNF a or TNF-α across all figures in the manuscript.

2) From the study~ AMPK pathway activation/deactivation is controlling downstream signaling event.

a) Is AMPK upstream signaling molecule in signaling pathway in PI3K/mTOR pathway?

b) Impact of in vitro and in vivo silencing either using inhibitor or siRNA of AMPK will affect downstream signaling events with and without metformin.

3) Did authors looked at the transcription factors induced by metformin at enhancer/promoter region in B cell, which id driving force for transactivating gene for higher production of TNF-α?

---

## [Author Response]

Essential revisions:Reviewer #1 (Recommendations for the authors):Better final culmination of data possible final diagram/schematic of pathway leading to TNF-α elevations in PCOS B cells, downstream effects and impact of Metformin.

Thank you for your valuable comments. We have modified the schematic shown in Figure 9.

Reviewer #2 (Recommendations for the authors):The authors state the role of TNF-α in PCOS is mainly driven by mTOR pathway, validated that level of TNF-α is contributed mainly by Bcells. Authors further showed the molecular mechanism involved in Bcells when exposed to metformin drug. Overall, the study is well written, and the finding is important for a logical design of immunotherapeutic strategy for PCOS. Following comments are to improve the manuscript.General Issues:1) Kindly keep same notion as TNF a or TNF-α across all figures in the manuscript.

We have kept same notion as TNF-α across all figures in the manuscript.

2) From the study~ AMPK pathway activation/deactivation is controlling downstream signaling event.a) Is AMPK upstream signaling molecule in signaling pathway in PI3K/mTOR pathway?b) Impact of in vitro and in vivo silencing either using inhibitor or siRNA of AMPK will affect downstream signaling events with and without metformin.

a) AMPK is not an upstream signaling molecule of the PI3K/mTOR pathway regulated by metformin. Metformin can activate mTOR through AMPK signaling pathway and also PI3K signaling pathway. Several studies have shown that metformin inhibits mitochondrial complex I and thereby oxidative phosphorylation leading to an increased AMP:ATP ratio, causing a direct activation of AMPK. Activation of AMPK leads to further inhibition of mTOR (*Kulkarni et al.*, *2020; Singh et al.*, *2021*). Metformin induces the M2 macrophage polarization to accelerate the wound healing via regulating AMPK/mTOR signaling pathway (*Qing et al.*, *2019*). Furthermore, metformin exerts anti-aging effects by downregulating insulin /IGF1 signaling, leading to PI3K/mTOR inhibition independently of AMPK (*Kulkarni et al., 2020*).

Metformin also exerts anti-cancer properties by inhibiting the PI3K/AKT/mTOR pathway (*Singh et al.*, *2021*).

b) Thanks for reviewer’s suggestion. In our study, metformin significantly increased the phosphorylation level of AMPK in pathological B cells from PCOS patients and mouse model. To further determine whether metformin inhibited TNF-α production in pathological B cells via regulating AMPK pathway, we studied the effect of dorsomorphin (compound C), an AMPK inhibitor, on TNF-α expression in pathological B cells in vitro. B cells isolated from peripheral blood of women with PCOS were cultured in RPMI-1640 medium containing B cell-targeted activator and metformin, with different concentrations of dorsomorphin (1μm, 2μm, 5μm) for 48hr. As shown in Author response image 1, dorsomorphin did not reverse the inhibitory effect of metformin on TNF-α production in pathological B cells. These results suggest that metformin's effect on B cells is accompanied by changes in AMPK phosphorylation levels, but AMPK is not required for metformin to inhibit TNF-α production in pathological B cells in our experiments. Similar to our findings, metformin-treated antigen receptor activated T cells show reduced glucose uptake compared to control antigen-activated cells, and metformin treatment blocks T cell antigen receptor-induced expression of c-Myc, HIF-1α and Glut1 in T cells. However, these inhibitory effects of metformin on T cells are independent of AMPK expression in T cells (*Zarrouk et al., 2014*). Next, we will further investigate the regulation mechanism of metformin on mTOR signaling in TNF-α-producing pathological B cells.

**Author response image 1. sa2fig1:** AMPK inhibitor dorsomorphin did not reverse the inhibitory effect of metformin on TNF-α production in pathological B cells from women with PCOS.

3) Did authors looked at the transcription factors induced by metformin at enhancer/promoter region in B cell, which id driving force for transactivating gene for higher production of TNF-α?

TNF-α is often described in the literature as one of the NF-κB-dependent proinflammatory cytokines. Chromatin immunoprecipitation (ChIP) assays revealed that metformin reversed the LPS-induced binding ability of NF-κB to both TNF-α and IL1β promoter in bovine mammary epithelial cells (*Xu et al.*, *2021*). In murine macrophages (RAW264.7 cells), ATF-3 binding to the promoters of TNF-α and IL-6 is enriched in the presence of metformin as compared with LPS alone, whereas LPS-induced NF-κB-binding to TNF-α and IL-6 promoter is reduced by metformin, suggesting that metformin-induced ATF-3 appears to suppress proinflammatory cytokine production via competition with NF-κB for binding to TNF-α and IL-6 promoters (Kim et al., 2014). These results suggest that NF-κB inactivation is associated with the inhibition of proinflammatory genes transcription by metformin. NF-κB plays critical roles in B cell development, activation and function (Sasaki et al., 2016). Upon TLR2 activation, the purified splenic B220^+^ B cells activate NF-κB to induce TNF-α mRNA expression (*Huang et al.*, *2012*). Therefore, we speculate that metformin may reduce TNF-α expression in B cells by reducing the binding ability of NF-κB to the TNF-α promoter region.

References

Huang, Z. M., Kang, J. K., Chen, C. Y., Tseng, T. H., Chang, C. W., Chang, Y. C.,... Leu, C. M. (2012).

Decoy receptor 3 suppresses TLR2-mediated B cell activation by targeting NF-kappaB. J Immunol, 188(12), 5867-5876. doi: 10.4049/jimmunol.1102516

Kim, J., Kwak, H. J., Cha, J. Y., Jeong, Y. S., Rhee, S. D., Kim, K. R., & Cheon, H. G. (2014). Metformin suppresses lipopolysaccharide (LPS)-induced inflammatory response in murine macrophages via activating transcription factor-3 (ATF-3) induction. J Biol Chem, 289(33), 23246-23255. doi: 10.1074/jbc.M114.577908

Kulkarni, A. S., Gubbi, S., & Barzilai, N. (2020). Benefits of Metformin in Attenuating the Hallmarks of Aging. Cell Metab, 32(1), 15-30. doi: 10.1016/j.cmet.2020.04.001

Qing, L., Fu, J., Wu, P., Zhou, Z., Yu, F., & Tang, J. (2019). Metformin induces the M2 macrophage polarization to accelerate the wound healing via regulating AMPK/mTOR/NLRP3 inflammasome singling pathway. Am J Transl Res, 11(2), 655-668.

Sasaki, Y., & Iwai, K. (2016). Roles of the NF-kappaB Pathway in B-Lymphocyte Biology. Curr Top Microbiol Immunol, 393, 177-209. doi: 10.1007/82_2015_479

Singh, S. K., Apata, T., Singh, S., McFadden, M., & Singh, R. (2021). Clinical Implication of Metformin in Relation to Diabetes Mellitus and Ovarian Cancer. Biomedicines, 9(8). doi:

10.3390/biomedicines9081020

Xu, T., Wu, X., Lu, X., Liang, Y., Mao, Y., Loor, J. J., & Yang, Z. (2021). Metformin activated AMPK signaling contributes to the alleviation of LPS-induced inflammatory responses in bovine mammary epithelial cells. BMC Vet Res, 17(1), 97. doi: 10.1186/s12917-021-02797-x

Zarrouk, M., Finlay, D. K., Foretz, M., Viollet, B., & Cantrell, D. A. (2014). Adenosine-monophosphate-activated protein kinase-independent effects of metformin in T cells. PLoS One, 9(9), e106710. doi: 10.1371/journal.pone.0106710